# Reduced-Parameter YOLO-like Object Detector Oriented to Resource-Constrained Platform

**DOI:** 10.3390/s23073510

**Published:** 2023-03-27

**Authors:** Xianbin Zheng, Tian He

**Affiliations:** College of Mechanical and Electrical Engineering, Qingdao University, Qingdao 266071, China; 2020020422@qdu.edu.cn

**Keywords:** neural network accelerator, object detection, FPGA, QNN

## Abstract

Deep learning-based target detectors are in demand for a wide range of applications, often in areas such as robotics and the automotive industry. The high computational requirements of deep learning severely limit its ability to be deployed on resource-constrained and energy-first devices. To address this problem, we propose a class YOLO target detection algorithm and deploy it to an FPGA platform. Based on the FPGA platform, we can make full use of its computational features of parallel computing, and the computational units such as convolution, pooling and Concat layers in the model can be accelerated for inference.To enable our algorithm to run efficiently on FPGAs, we quantized the model and wrote the corresponding hardware operators based on the model units. The proposed object detection accelerator has been implemented and verified on the Xilinx ZYNQ platform. Experimental results show that the detection accuracy of the algorithm model is comparable to that of common algorithms, and the power consumption is much lower than that of the CPU and GPU. After deployment, the accelerator has a fast inference speed and is suitable for deployment on mobile devices to detect the surrounding environment.

## 1. Introduction

In the composition of the robot system, an excellent and reliable vision system not only gives the robot basic functions such as target judgment and distance estimation but also provides the control system with more accurate information about environmental variables so that the robot can accurately judge its surroundings and provide better control information [1]. Perception is one of the more important components of robot intelligence, and most perception today consists of a visual component as the main part and sensors to assist vision. In robot vision, target detection and image segmentation are mainly responsible for determining the orientation and size of the inspected items [2]. For example, in the target grasping task, the target needs to be detected first, such as by using Fast RCNN [3] and YOLO [4], to determine the location and distinguish the detected object in the image and determine the class of the target, or Mask RCNN [5] can be used to directly determine the location and class of the object while segmenting the target and determining its shape.In the case of binocular vision, the distance between the measured object and the robot can be calculated by geometrical principles through the distance between different visual sensors and the distance between the sensor and the measured target. If a sensor such as LIDAR is used, a point map can be directly generated to calculate the distance to the target [6].

However, at this stage, vision systems based on deep neural networks usually need to be deployed on a general-purpose computing platform such as a GPU. Although GPUs have huge advantages over other platforms in the training phase, there are several problems with deploying GPUs into robotic systems [7]. First of all, the GPU is very bulky and requires an excellent design to deploy it into a robot system, but usually miniaturized robots cannot meet its minimum size requirements. Secondly, the GPU consumes a great deal of power, and the battery power of small robots cannot be increased due to the limitation of the load. Thirdly, GPUs generate a great deal of heat when running, so they require good cooling devices (usually fans for air cooling) and cannot be added very efficiently to certain situations with lenient environmental requirements (such as dust and humidity). For a highly integrated robotic system, it is necessary for practical applications to design an efficient target detection algorithm and deploy it in a hardware accelerator [8].

Starting from the above problem, we choose FPGA as the hardware gas pedal of the algorithm [9]. FPGA is a semi-custom circuit, which is a logic array that can be programmed. FPGA has a shorter design cycle and lower cost than CPU and ASIC, which are logic-fixed arrays, and using the parallel computing feature of FPGA, we can quickly perform inference calculations on neural networks, so FPGA is the best platform for deep learning model deployment acceleration [10,11].

For the accelerator, it is finally to be deployed in the actual application environment, which is an important research topic for hardware accelerators. Prior to this, there have been many studies on the practical application of neural network accelerators [12,13,14,15] and the design of neural network accelerators [16,17,18]. However, most of the existing algorithm models are transformed and arranged on the FPGA, and the model design is not combined with the hardware architecture.Therefore, in this work, we design a YOLO algorithm and deploy the algorithm model to run on FPGAs. This work has two main contributions:Considering the limited resources on the FPGA chip when designing the model algorithm, the following methods are used in the algorithm model to reduce the model size and computational effort: (1) reduce the hardware computation and number of parameters using Depthwise Separable Convolution [19] and network quantization [20]. (2) Improve network accuracy by adding SE modules [21] within specific network layers. (3) A single lightweight detection head is designed to improve the running speed of the algorithm. (4) Improving network training quality with dynamic positive and negative sample assignment, data augmentation, and multiple candidate targets across grids [22,23].On the FPGA side, based on the Xilinx FINN framework [24,25], the neural network written by Pytorch [26] is converted to ONNX format, and the basic operators are written by HLS. The neural network nodes in ONNX format are transformed into hardware code with custom operators via HLS, resulting in a low latency, low power, and highly accurate neural network.

## 2. Algorithm Design

### 2.1. Overall Algorithm Model Design

To minimize the consumption of FPGA on-chip resources, we do not use a mature target detection algorithm such as YOLOv3 [27]. Instead, we redesigned a deep neural network based on the YOLO algorithm for training and deployment to FPGAs. This network algorithm model is mainly divided into two parts, which are the backbone network structure design and detection head network structure design.

#### 2.1.1. Backbone Design

The basic part of the backbone network is derived from the Resnet-18 [28] network structure. The 3 × 3 ordinary convolution kernel in the original Resnet-18 network is changed to a Depthwise Separable Convolution. After this change, the accuracy is almost lost, and the number of parameters is greatly reduced. Depthwise Separable Convolution was first introduced by Google in the inception of GoogleNet v3 [29] and was first mainly applied in MobileNetv1 [19]. Depthwise Separable Convolution is different from standard convolution, which acts on the entire channel of the input feature map, while Depthwise Separable Convolution splits this operation into two parts, i.e., Depthwise Convoultion and Pointwise Convolution.
(1)$$Kernel\;Size\;=H_{k}\times W_{k}\;$$
(2)$${{Filter\;Parameter\;=C}_{out}\times H_{k}\times W_{k}\times C}_{in}$$

Suppose that an input feature map is 5 × 5 × 3 (H, W, C), the output feature map is 5 × 5× (H, W, C), the shape of the convolution kernel is 3 × 3 (padding = 1), and the filters parameter is calculated by Equation (Equation 2) as 4 × 3 × 3 × 3 = 108, shown in Figure 1. Usually, ordinary convolution is a convolution kernel responsible for all channels of the overall input image, while Depthwise Convolution is a convolution kernel responsible for one channel of the input image. Output the feature map of each channel after convolution, and the number of output channels is the same as the number of input channels is equal, as shown in Figure 2. Then, assume that an input feature map is 5 × 5 × 3 (H, W, C), and the output feature map is 5 × 5 × 4 (H, W, C); after Depthwise Convolution, the size of the output feature map is still 5 × 5 × 3, the size of the convolution kernel is 3 × 3, and the number of parameters by Equation (Equation 2) is 1 × 3 × 3 × 3 = 27. Next, the Pointwise Convolution operation is very similar to the regular convolution, but its convolution kernel size is 1 × 1. Such a convolution operation will combine the feature map output from the Depthwise Convolution in the depth direction for weighting and generate a new feature map, shown in Figure 2. According to the above assumptions, if the output feature map is 5 × 5 × 4 (H, W, C), the number of filter parameters is 4 × 1 × 1 × 3 = 12, and the total number of parameters is 27 + 12 = 39. The total number of parameters of the Depthwise Separable Convolution is only about 40% of the normal convolution, which can significantly reduce the number of parameters and computation of the convolution operation.

In order to increase the image features that can be extracted by the backbone, we add the quantized SE module [21] to the network. The SE module was changed, and the original FC layers were replaced with a 1 × 1 convolution operation as shown in Figure 3.

This change from the original full FC layer design avoids the dimensional conversion between the input as a matrix and output as a vector. We changed and added the above modules to the original ResNet-18 network and finally reduced the overall number of parameters from 11.7M to 1.05M. The structure of the overall backbone is shown in Figure 4.

#### 2.1.2. Detection Head Design

This network algorithm designs a lightweight single-detection head to predict the target type and bounding box. Before introducing our single-detection head, we need to introduce the multi-detection head. After YOLOv2 [30], YOLOv3 [27], YOLOv4 [31], and YOLOv5 pass the feature maps output by the backbone through three detection heads of different sizes and finally output three sets of prediction data of different sizes in parallel. This multi-detector head design can make good use of the different resolutions of the three detection heads to detect large, small, medium, and three kinds of targets. The design of the detection head of this network mainly refers to the design ideas of YOLOF [32] and TridentNet [33] and adopts a 5 × 5 grouped convolution parallel network structure similar to inception [29], shown in Figure 5. It is expected to fuse the features of different receptive fields so that the single detection head can adapt to objects of different scales.

### 2.2. Network Optimization Design

#### 2.2.1. Neural Network Quantization

Traditional neural network training and inference are carried out on the PC platform using GPU, as there is no shortage of resources and it supports floating-point operations. Therefore, the data format of the neural network trained by PC is usually a 32-bit floating-point number (FP32), which greatly increases the amount of data while obtaining high precision. Each bit of data occupies 32 bits of space. This drastically reduces the size of the deployable network on platforms with limited computing resources and requires the quantization of the network as many mobile computing platforms do not support floating point computing.

The network quantization part is based on Xilinx Brevitas, a Pytorch library for neural network quantization. By specifying the quantization type of the node (such as 8-bit integer (INT8) or others) during the model building period in the function provided, quantization training can be performed at the training model stage. Brevitas also provides an ONNX-based export method, which can export the trained model to the “.ONNX” format for better deployment of the model. This study mainly uses Brevitas to quantify the weight and activation of the network to 4 bits, and its comparison with other network model parameters is shown in Table 1.

#### 2.2.2. Data Format

The default data storage format of Pytorch is NCHW (N, channels, height, width), and in this design, the data storage format in the network is NHWC. As shown in Figure 6, RGB pixels stored in NCHW format are arranged in the outermost layer with the pixels in each channel next to each other, i.e., “RRRRRRRGGGGGBBBBB”, while RGB pixels stored in NHWC format are arranged in the innermost layer with the pixels in the corresponding spatial locations of multiple channels next to each other, i.e., “RGBRGBRGBRGBRGBRGB”.

Assuming that a color-to-grayscale calculation is needed for the feature map, the NCHW calculation process is shown in Figure 7. All the values of the R channel are multiplied by 0.3, all the values of the G channel by 0.55, all the values of the B channel by 0.15, and finally, the three channels are summed to obtain the grayscale value.

For NHWC format grayscale calculation, as shown in Figure 8, the first R pixel is multiplied by 0.3, the second G pixel by 0.55, and the third B pixel by 0.15, which are added to get the first grayscale pixel, and finally the above steps are repeated to get all the gray values. The computational complexity of the two data formats is the same. However, the NCHW format needs to occupy a large temporary space, and all channel data need to be saved to obtain the final result, while NHWC has a better locality, and can consume fewer resources to perform calculations in FPGA than NCHW. For convolution operations, if the convolution kernel is 1 × 1, each input channel is multiplied by a weight, and then all channel results are accumulated to obtain an output channel. If the NHWC format is used, the convolution calculation can be reduced to a matrix multiplication calculation, i.e., the 1 × 1 convolution kernel implements a linear transformation from each input pixel group to each output pixel group. This network structure uses a large amount of Depthwise Separable Convolution, and the pointwise convolution is a 1 × 1 convolution kernel to transform the output channel, so in this design, using the NHWC format as the data storage format can speed up the calculation speed and simplify the complexity of the operation.

## 3. Hardware Orient Algorithm Optimization

### 3.1. FPGA Operator Design

The neural network model trained on the PC side only supports inference and training on a general-purpose computing platform. The design needs to implement neural network inference acceleration on a custom FPGA platform, so it is necessary to write inference operators suitable for the FPGA platform. Through the FINN [24] framework, we can only write a general custom op and convert the custom op to an HLS file, and the framework realizes automatic routing in the synthesis stage by reading the model parameters of the ONNX format file. The FINN framework has built-in common custom operators such as the convolution input generator, Streamling Maxpool batch, and Matrix vector activation, which only support simple deep neural network transformations, while for this neural network algorithm, a custom-written custom op is required.

#### 3.1.1. Streamling Mul Op

Because the SE module [21] is used in the backbone of this algorithm model, the output of the SE module needs to be multiplied by the output vector of the SE module for each channel of the input data on the output of the corresponding tap. Thus, it is necessary to write the corresponding op for this node to complete the corresponding hardware operation.

The Streamling Mul Op is written based on Xilinx Vitis HLS using the Stream library in Vitis HLS to convert the input feature map into stream data, because the overall model uses NHWC format to transmit data, so the stream data format is as shown in Figure 9. The input and output data are manipulated through C++ code, and the input main road feature map (Resnet input) is multiplied by the SE input output by the SE module, and finally the output is produced.

As shown in Figure 10, the 0-channels data of the ResNet input are loaded through hls::stream and the 0-channels data of the SE input are loaded at the same time. The two data are multiplied and output to the output. This is cycled H*W times to complete the Streamling Mul steps.

#### 3.1.2. Streamling Concat Op

Because the detection head of this algorithm model needs to fuse the features of the three resolutions, the Concat module is required to accumulate the feature maps, so it is necessary to write the corresponding op for this node to complete the corresponding hardware operations. The input and output data are still in the NHWC format, which has been introduced above, see Figure 9, and its main operation process is shown in Figure 11.

The data of 0-channels of Input 1 are passed to the output through hls::stream, and then the data of 0-channels of Input 2 are passed to the output. At this time, the data of channels × 2 exist in the output and those of 0-channels are Input 1. Channels-channels × 2 is Input 2; this step is repeated H*W times, and the original N × H × W × C data of Input1 and Input2 are concatenated into N × H × W × (C × 2) data, completing the Concat operation.

### 3.2. General Framework

This target detection vision algorithm is deployed on the ZYNQ platform. Using the ZYNQ platform, we can use low-power, highly customized hardware to replace high-energy, large-space, and high-cost deep-learning workstations. As shown in Figure 12, the overall algorithm deployed on ZYNQ is divided into two parts, mainly composed of neural network accelerators, running on the PL (Programmable Logic) side, and the other part running on the PS (Processing System) side. It is mainly used to obtain input images and for general calculations, etc. The PS end and the PL end are connected to transmit data through AXI GPIO.

In the beginning, the camera acquires real-time image data at the PS side and preprocesses the image data. The image data are usually saved in NCHW format, which needs to be converted to NHWC format by the transpose operation and then sent to the PL side by the AXI bus for network inference, which is loaded with the compiled neural network gas pedal. After inference, the AXI bus transfers the results back to the PS side for classification, boundary box prediction, and other operations.

### 3.3. Deployment

The deployment side mainly includes two parts, the host side and the FPGA side, with the host side mainly performing software deployment operations and the FPGA side performing hardware deployment operations. The host side mainly relies on the Xilinx FINN [24] framework, whose core functions are shown in Figure 13 including the quantization neural network model, quantization training, streamline operation, conversion of generic nodes to custom op, conversion of the custom op to HLS C++ files, and synthesized HLS generation bitstream. Quantization is performed by Xilinx Brevitas, and its purpose is to convert 32-bit floating-point numbers into integers. The specific functions are carefully discussed in Section 2.2.1.

The core process of the streamline operation is to simplify the model through the pre-written streamline module. Through the streamline operation, some linear operations such as add, mul, etc. can be linearly transformed into the parameter of the node and integrated with nodes such as MultiThreshold.

Figure 14a shows the block after the streamline operation. After streamline, the complexity of the network model is greatly reduced. Most linear operations are integrated into MultiThreshold to prepare for the generation of HLS C++ files.

Converting generic nodes to custom op, converting custom op to HLS C++ files, and synthesizing HLS to generate a bitstream are three types of operations that can be categorized together as compilation operations. The general nodes (such as Conv, MultiThreshold, etc.) are converted in the streamlined model to custom op (such as Conv into a ConvolutionInputGenerator). After converting all nodes into custom op nodes, as shown in Figure 14b, the pre-designed function of the FINN framework is used to sequentially convert custom op into HLS files, and Vivado synthesizes the IP core (intellectual property core) created by HLS to generate the bitstream file. After the above steps, the deep neural network model file that ZYNQ can recognize is generated, and the file is finally loaded into the PL of ZYNQ. Using PYNQ (a Python implementation based on the ZYNQ platform), the bitstream file can be loaded in real time to call the network model and accelerate the reasoning of the neural network model to realize the neural network accelerator.

## 4. Experimental Results

### 4.1. Experimental Design

The experiment is divided into model building, training, quantization, and synthesis on the host side and image acquisition and inference on the hardware side. The network model is built and trained on the host side, as well as verifying whether it meets the hardware requirements. Quantification and synthesis are based on the Xilinx FINN framework, HLS file generation using Vitis HLS 2022.1, and synthesis and layout using Vivado 2022.1. The hardware platform is a custom hardware experimental board with Xilinx ZYNQ UltraScale+ MPSoCs XCZU7EV–2FFVC1156I, a camera with Logitech C270, and an experimental board with PYNQ system for general-purpose program processing.

### 4.2. Host Side

On the host side, our model was mainly used for training, on a GTX3090 equipped workstation, with a maximum Epoch initially designed for 300. In the training process, the training set and validation set data are mainly from the COCO dataset [34], totaling 20.1 GB, of which the training set contains 118,187 images, totaling 19.3 GB, and the validation set contains 5000 images, totaling 0.8 GB. The training process takes a long time. A round of Epoch takes nearly 20 min. After 300 Epoch training, map@0.5 is 39.4%. Although mAP is lower than target detectors such as YOLOv5s, we tested the inference speed of our network and other algorithms on the same workstation, and the inference time was 4.5 ms/1 fps. The comparison with other algorithms is shown in Table 2.

### 4.3. FPGA Side

The efficiency and accuracy of this network on the host side shows that our approach is fully capable of handling the target detection requirements in most cases, so the trained neural network is ported to the FPGA side. The FINN environment is deployed using Docker, and the trained network is quantized, streamline operated, HLS generated, and hardware synthesized; then, the synthesized neural network file is deployed to the PL side of ZYNQ to perform inference, which completes the porting and deployment of the neural network from the PC side to the FPGA side. After combining the networks, the overall network requires 105,372 LUT, 158,842 FF, 193.5 BRAM, and a total on-chip power of 5.508 W’, which is a significant advantage compared to the PC side.

### 4.4. Results

The network model achieves better results on the COCO 2017 dataset. The model not only has good classification accuracy but also has good inference speed, and the model can be deployed to the FPGA side for acceleration, which greatly reduces the power used to run the model, with good timing and energy consumption ratio. In this study, for the first time, the YOLO-like network model is integrated and deployed to FPGA hardware through the FINN framework. The hardware solution is relatively moderate in price, relatively easy to deploy, and highly stable with high design flexibility, which is suitable for most robotic systems with limited hardware size.

### 4.5. Further Discussion and Analysis

Quantization accuracy: This network was trained and validated using FP32 accuracy in the validation phase, and the accuracy was 95.1% at the end of the training, while the accuracy of the network decreased by 8% after quantizing the network to 4 bits. However, after quantization, the size of the network model is greatly reduced, which is suitable for synthesis and deployment to the FPGA side. We believe that it is a good solution to exchange a smaller accuracy loss for a larger resource overhead by quantizing the model in a resource-constrained environment.

Mathematical model analysis: The construction of this algorithm model is a black box, without mathematical analysis. In future research, it is necessary to construct its mathematical model, analyze it from a mathematical point of view, and use mathematical tools [37,38] to improve the accuracy and speed of the algorithm.

Performance improvement at the FPGA side: On the host side, the input image has little impact on the efficiency of the overall detection process. However, at the ZYNQ side, since there is no hardware interface between PS and PL for communication, and the image input and preprocessing are not based on programmable hardware circuits, the efficiency is limited by the performance of the ARM core at the PS side, so the overall detection efficiency is limited by the performance at the PS side. If the image preprocessing is also made into a programmable hardware circuit, the overall detection performance should be significantly improved.

Practical application: This target detection accelerator is only used for simple object detection at this stage, but it has many practical application scenarios. Longzhen Yu et al. used FPGA to build a defect detector to achieve good accuracy and speed in industrial inspection [39]. Jiaqi Zhai et al. used FPGA to build a license plate detector [10], and this detector has great application prospects in daily practice.

## 5. Conclusions

We implemented a quantized YOLO-like target detection algorithm and ported it to an FPGA platform for inference acceleration. In terms of the algorithmic model, we used Depthwise Separable Convolution, lightweight single detection head design, and model quantization to significantly reduce the model size and the amount of calculation. In terms of hardware design, we designed ops dedicated to this network algorithm through the FINN framework, converted the network model into an FPGA accelerator, and realized a target detection accelerator with a power consumption of 5.508 W. The target detection accelerator can be widely used in situations requiring low power consumption and high real-time performance, such as for robots.

There are still many directions worthy of research in neural network accelerators. In the future, a specific neural network can be used to perform more detailed processing on model quantization so that each node can perform different quantization operations. This operation can optimize the consumption of on-chip resources when converting the model into a hardware structure, which can be used as a research direction for the establishment and deployment of neural network accelerator models in the future.

## Figures and Tables

**Figure 1 sensors-23-03510-f001:**
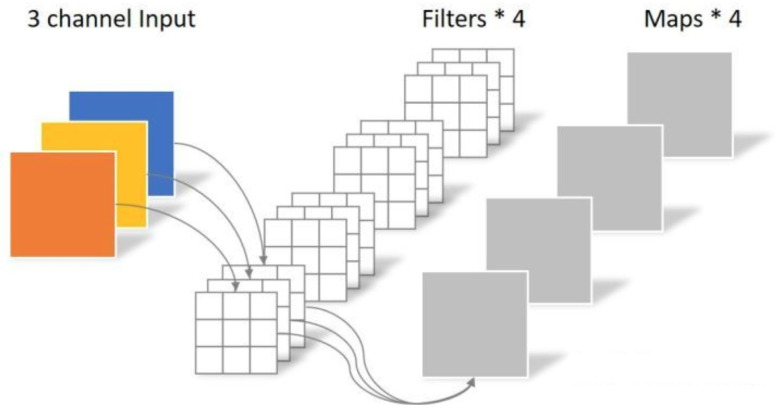
Convolution process illustration.

**Figure 2 sensors-23-03510-f002:**
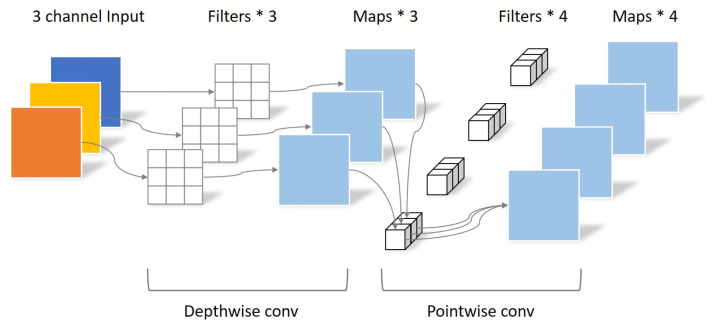
Depthwise Separable Convolution process illustration.

**Figure 3 sensors-23-03510-f003:**
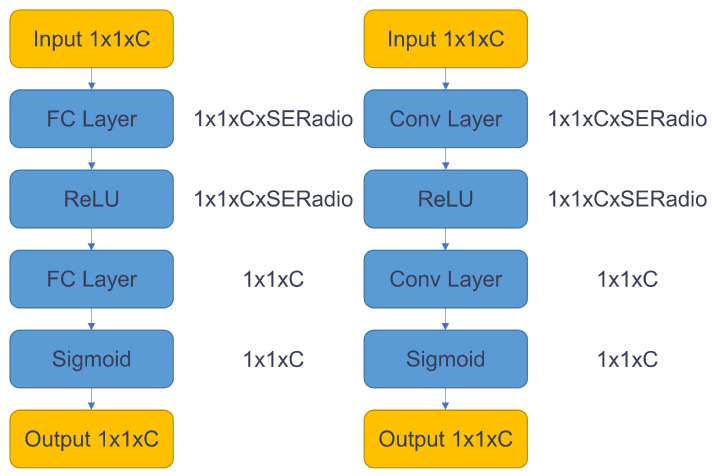
Use of 1 × 1 convolution to replace FC layers in SE module.

**Figure 4 sensors-23-03510-f004:**
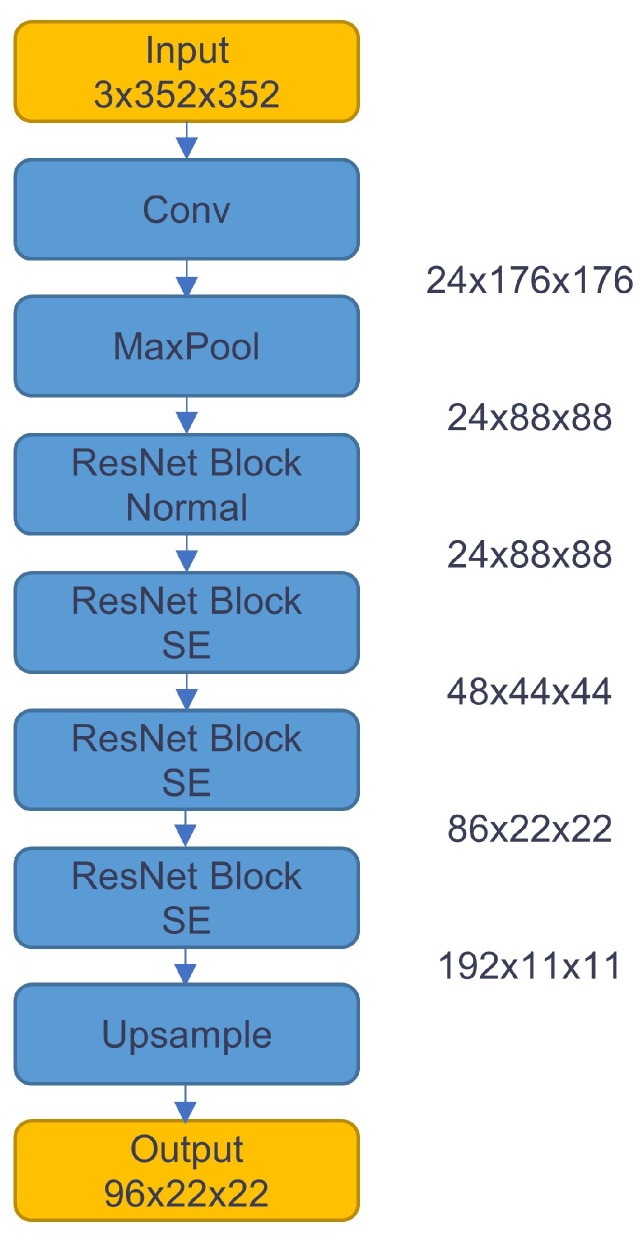
Overall backbone structure.

**Figure 5 sensors-23-03510-f005:**
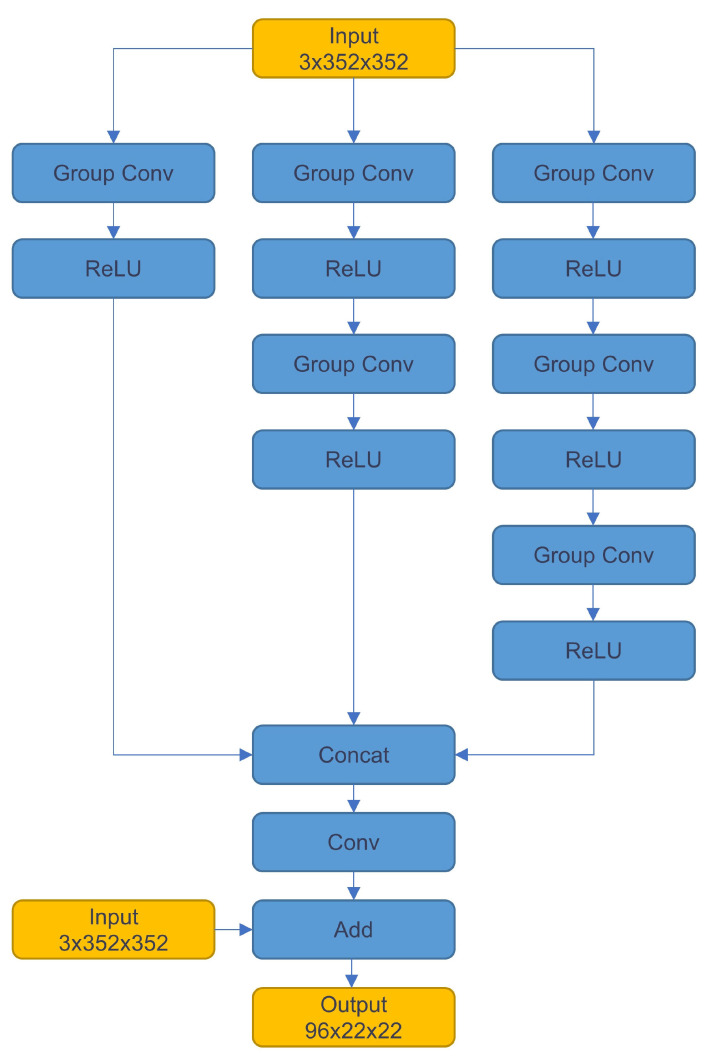
Detection head multi-channel fusion structure.

**Figure 6 sensors-23-03510-f006:**
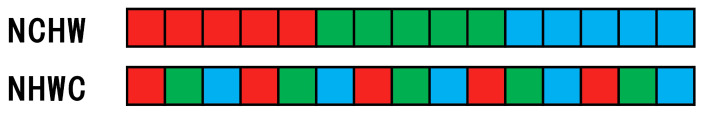
NCHW and NHWC format.

**Figure 7 sensors-23-03510-f007:**
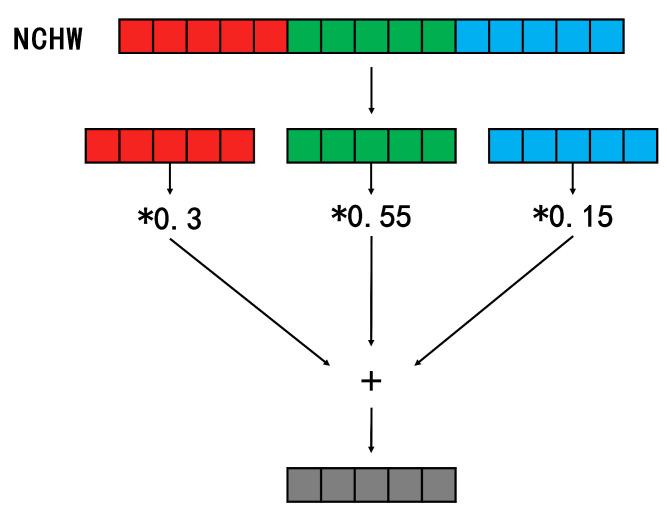
Calculation of grayscale in NCHW format.

**Figure 8 sensors-23-03510-f008:**
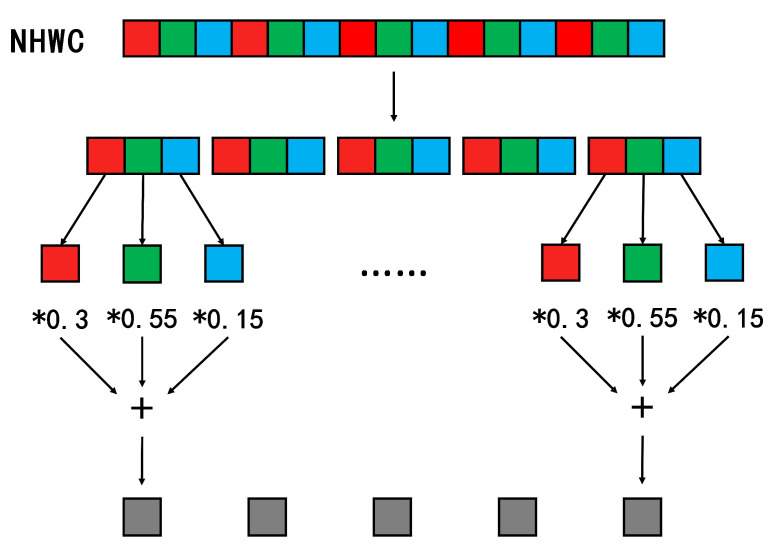
Calculation of grayscale in NHWC format.

**Figure 9 sensors-23-03510-f009:**
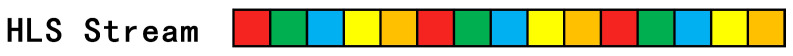
HLS stream data.

**Figure 10 sensors-23-03510-f010:**
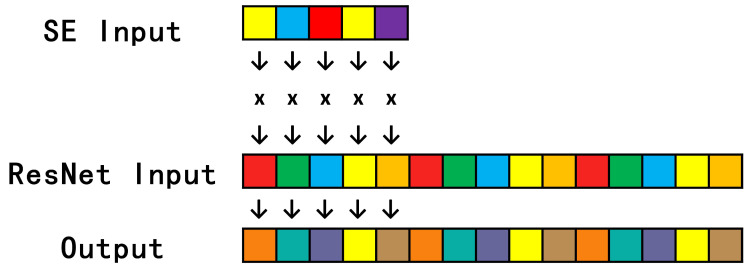
Stream Mul process.

**Figure 11 sensors-23-03510-f011:**
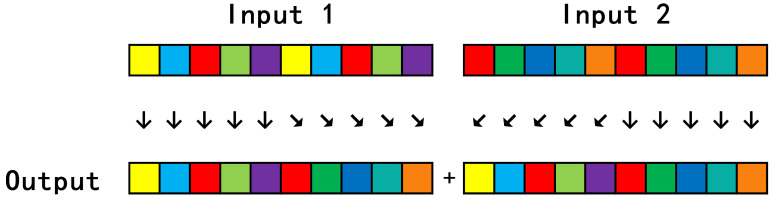
Stream ConCat process.

**Figure 12 sensors-23-03510-f012:**
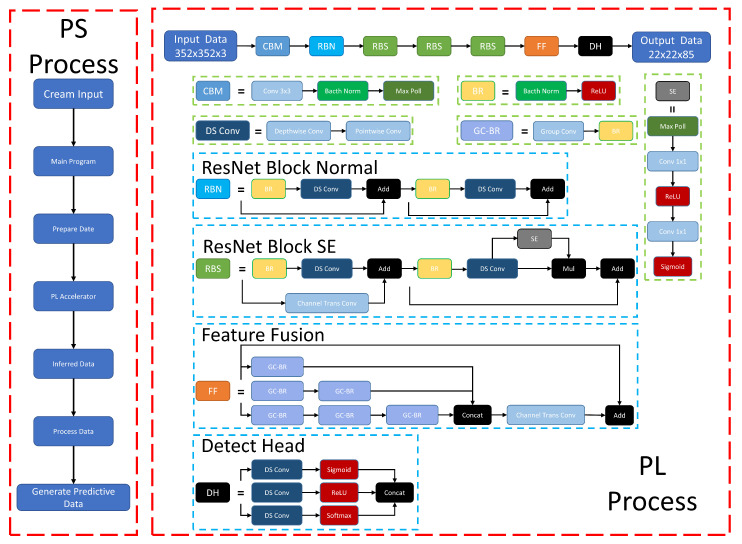
Overall framework of the Target Detection Accelerator.

**Figure 13 sensors-23-03510-f013:**
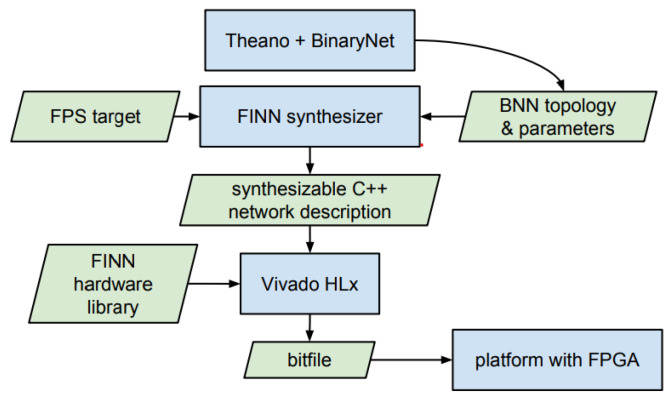
FINN core process.

**Figure 14 sensors-23-03510-f014:**
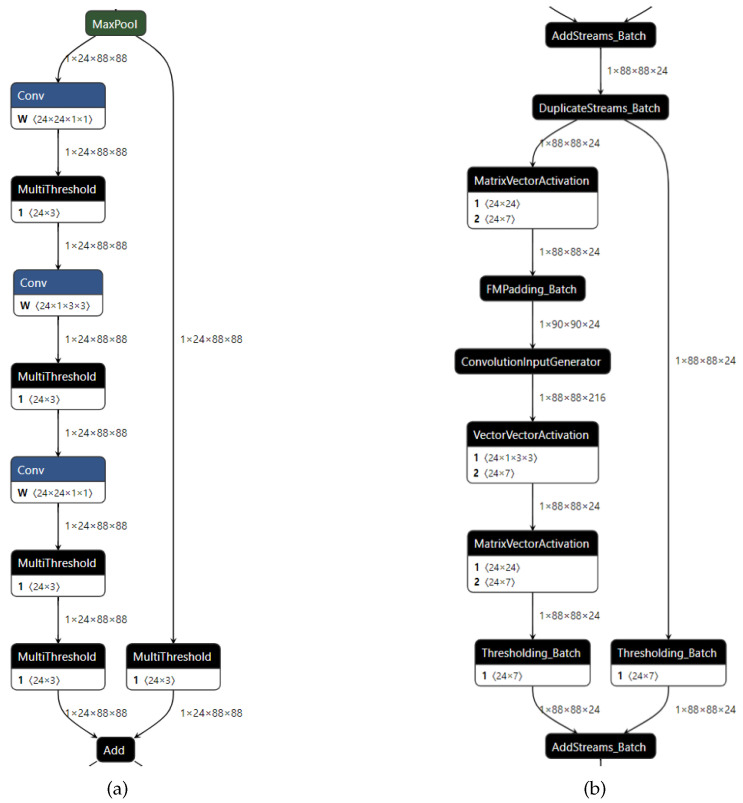
Two-step conversion operation in the FINN framework. (**a**) The network transformed by the streamline step. (**b**) The network transformed by the conversion to HLS step.

**Table 1 sensors-23-03510-t001:** Comparison with other model sizes.

Model Name	Model Size	Data Type
YOLOv3-Tiny	33.7 MB	FP32
YOLOv4-Tiny	23.0 MB	FP32
YOLOv5s	27.8 MB	FP32
Our Net	9.4 MB	INT4

**Table 2 sensors-23-03510-t002:** Comparison with other algorithms.

Model Name	Params (M)	Speed (ms)
Yolov5s	7.2	8.8
Yolov6s [35]	17.2	10.1
Yolov7-tiny [36]	6.01	12.8
Our Net	0.9	4.5

## Data Availability

Not applicable.

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
