# Peer review of "Reduced-Parameter YOLO-like Object Detector Oriented to Resource-Constrained Platform"

_sensors, 2023, doi:10.3390/s23073510_

Round 1
Reviewer 1 Report
The manuscript is interesting but I would suggest the introduction and discussion part to be supported more with the relevant literature about the potential practical applications of this algorithm in detecting surrounding environment.
The abstract and conclusion part should also emphasize these practical applications more.
Author Response
The manuscript is interesting but I would suggest the introduction and discussion part to be supported more with the relevant literature about the potential practical applications of this algorithm in detecting surrounding environment.
The abstract and conclusion part should also emphasize these practical applications more.
Response: Practical applications of this accelerator have been added to the Introduction section and Subsection 4.5 further discussion and analysis (In page 2, line 50 and page 13, line 341.).
Reviewer 2 Report
Acceptable in current form. The Paper is of good technical impact.
Author Response
Acceptable in current form. The Paper is of good technical impact.
Response: The authors are grateful for the reviewer’s approval of the manuscript.
Reviewer 3 Report
Dear authors,
I have the honour to review your paper. The comments is attached.
Best wishes,
the reviewer

Author Response
1.page 2, line 50, in ”There have been many studies focusing on accelerator design for neural networks[12].” The author said ”many studies”, however, there is only one reference.
Response: Two more relevant references have been added accordingly (In page 2, line 54.).
2.page 2, line 80, in “...Depthwise Separable Convolution.After this change,” Full point “.” and two words are closely connected. This situation occurs in many places in the paper. This requires careful examination and revision of the full text.
Response: All the typos mentioned have been corrected across the manuscript.
- page 2, line 85–86, “...splits this operation into two parts. That is, Depthwise Convoultion and Pointwise Convolution.” should be “...splits this operation into two parts, i.e., Depthwise Convoultion and Pointwise Convolution.”
Response: All the mentioned textual problems have been fixed.
- page 3, line 94, “...the same as The number of...” should be “...the same as the number of...”; “...channels is equal, shown...” should be “...channels is equal, as shown...”.
Response: All the mentioned textual errors have been corrected.
- page 3, line 109, “...convolution operation. shown in Figure 3.” should be “...convolution operation. As shown in Figure 3.”
Response: The mentioned textual error has been corrected.
- page 4, line 117, “After YOLOv2[24]. YOLOv3[21], YOLOv4[25]” should be “After YOLOv2[24], YOLOv3[21], YOLOv4[25]”.
Response: The mentioned typo has been corrected.
- page 6, line 157, “then repeat The above steps” should be ”then repeat the above steps”.
Response: The mentioned typo has been corrected.
- page 8, line 187, in “This OP is written...”, Is “OP” the “Op” in “Streamling Mul
Op”? If yes, their writing should be consistent.
Response: The “OP” has been changed to “Streamling Mul Op” to make the expression consistent (In page 8, line 190.).
- The description models in this paper are discrete. It is suggested that the author add and refer to the following references [1] and [2], and establish a continuous model to study this problem in the future.
Response: A discussion on mathematical models has been added to Subsection 4.5 further discussion and analysis with the recommended references cited (In page 12, line 302.).
Round 2
Reviewer 1 Report
The manuscript has been sufficiently improved.
Reviewer 3 Report
Dear authors,
The author has made modifications based on the review suggestions.
I recommend that the paper be published in Sensors.
Best regards,
the reviewer